# *Hops/Tmub1* Heterozygous Mouse Shows Haploinsufficiency Effect in Influencing p53-Mediated Apoptosis

**DOI:** 10.3390/ijms22137186

**Published:** 2021-07-02

**Authors:** Simona Ferracchiato, Nicola Di-Iacovo, Damiano Scopetti, Danilo Piobbico, Marilena Castelli, Stefania Pieroni, Marco Gargaro, Giorgia Manni, Stefano Brancorsini, Maria Agnese Della-Fazia, Giuseppe Servillo

**Affiliations:** Department of Medicine and Surgery, University of Perugia, Piazzale L. Severi 1, 06129 Perugia, Italy; simona.ferracchiato@unipg.it (S.F.); nicola.diiacovo@studenti.unipg.it (N.D.-I.); damiano.scopetti@studenti.unipg.it (D.S.); danilo.piobbico@collaboratori.unipg.it (D.P.); marilena.castelli@collaboratori.unipg.it (M.C.); stefania.pieroni@unipg.it (S.P.); marco.gargaro@unipg.it (M.G.); giorgia.manni@unipg.it (G.M.); stefano.brancorsini@unipg.it (S.B.)

**Keywords:** *Hops/Tmub1*, heterozygous mouse, p53, apoptosis, DNA-damage response, haploinsufficiency

## Abstract

HOPS is a ubiquitin-like protein implicated in many aspects of cellular function including the regulation of mitotic activity, proliferation, and cellular stress responses. In this study, we focused on the complex relationship between HOPS and the tumor suppressor p53, investigating both transcriptional and non-transcriptional p53 responses. Here, we demonstrated that *Hops* heterozygous mice and mouse embryonic fibroblasts exhibit an impaired DNA-damage response to etoposide-induced double-strand breaks when compared to wild-type genes. Specifically, alterations in HOPS levels caused significant defects in the induction of apoptosis, including a reduction in p53 protein level and percentage of apoptotic cells. We also analyzed the effect of reduced HOPS levels on the DNA-damage response by examining the transcript profiles of p53-dependent genes, showing a suggestive deregulation of the mRNA levels for a number of p53-dependent genes. Taken together, these results show an interesting haploinsufficiency effect mediated by *Hops* monoallelic deletion, which appears to be enough to destabilize the p53 protein and its functions. Finally, these data indicate a novel role for *Hops* as a tumor-suppressor gene in DNA damage repair in mammalian cells.

## 1. Introduction

Hepatocyte Odd Protein Shuttling (HOPS), also known as Trans-Membrane and Ubiquitin-like protein-1 (TMUB-1)—hereafter referred to as HOPS—is a ubiquitin-like protein that acts as a modifier in the control of several cell functions such as centrosome assembly, proliferation, inflammation, and apoptosis [1,2,3,4]. The *Hops* gene is expressed in all organs examined to date, translating into three different proteins with distinct molecular weights of 27, 24, and 21 kDa [1]. Structural analysis of HOPS revealed a ubiquitin-like domain (UBL), a proline-rich region, and three transmembrane regions [1]. After mitogenic stimuli such as surgical partial hepatectomy, cAMP or growth-factor treatment, oxidative stress, or DNA damage, HOPS moves from the nucleus to the cytoplasm via the exportin CRM1 (Chromosomal Maintenance 1, also known as Exportin 1) [2,3,4,5]. HOPS is an essential protein for centrosome assembly; its knockdown results in altered mitosis characterized by multipolar spindles and mis-segregation of DNA, which in turn activate p53 checkpoint and cell cycle arrest [4]. In proliferation, it has been shown that the increased level of IL-6, a fundamental cytokine in priming liver regeneration, activates C/EBPβ, which in turn upregulates HOPS expression [6,7]. Moreover, during liver regeneration, HOPS inhibits STAT3 pathways, negatively controlling hepatocyte proliferation [8]. HOPS negatively correlates with hepatocellular carcinoma malignancy, inhibiting proliferation via STAT1 signaling and promoting ubiquitination of ΔNp63 [9]. Analyzing HOPS’ role in hepatocellular carcinoma proliferation, it has been shown that tumor cell growth is suppressed by controlling the ubiquitination and degradation of ΔNp63 isoforms, thus driving cells to apoptosis [10]. HOPS has been identified as an important regulator of the p19Arf–NPM (nucleophosmin) complex in the nucleolus [3]. In controlling p19Arf stabilization, HOPS retains MDM2 (Mouse Double Minute 2)—the main negative p53 regulator—and controls p53 half-life [3]. This complex circuit is essential for maintaining and regulating intracellular levels and p53 activity, since many tumors can arise due to alterations affecting these circuits. Recently, it has been demonstrated that HOPS is directly involved in p53 stabilization, controlling the p53-mediated mitochondrial apoptosis response and p53 nuclear import [11]. HOPS, through its UBL, acts as a modifier to control p53 stability, inhibiting ubiquitination and sustaining the cytoplasmic concentration of p53 to trigger mitochondrial apoptosis. [11]. The role of HOPS as a modifier has also been demonstrated in controlling the activation of NF-kB pathway, regulating TRAF-6 stability and, in turn, modulating NF-kB response to LPS. Lack of HOPS is associated with a reduction in inflammatory response after treatment [12].

The tumor suppressor p53 represents one of the most studied proteins in the medical and biological fields, playing a pivotal role in protecting cells from malignant transformation by inducing cell cycle arrest or apoptosis [13,14,15,16,17]. The importance of the TP53 gene in human cancer progression is highlighted by the fact that its alterations are displayed in over half of all human cancers [14,18,19]. The p53 protein plays an important role in DNA-damage-induced apoptosis, partly by acting as a transcription factor to direct the expression of apoptotic mediators. Moreover, a significant amount of p53 accumulates in the cytoplasm, inducing apoptotic events by acting directly at mitochondria. In response to genotoxic stresses, p53 is phosphorylated at serine 15 by both the ATM/ATR (Ataxia Telangiectasia Mutated/Ataxia Telangiectasia and Rad3-Related Protein) protein kinases [20,21], preventing its interaction with its negative regulator MDM2 and, thus, its degradation [22,23]. Therefore, stabilized p53 accumulates in the nucleus, binding to specific DNA sequences and upregulating the transcription of several pro-apoptotic members of the BCL-2 family, such as BAX (BCL2 Associated X Protein), PUMA (P53 Up-Regulated Modulator Of Apoptosis) and NOXA (Phorbol-12-Myristate-13-Acetate-Induced Protein 1) [24,25,26], which results in the activation of the apoptotic cascade [27].

Due to HOPS’ involvement in several cell functions, we were interested in exploring the molecular significance underpinning the HOPS heterozygote configuration in promoting apoptosis. Indeed, apoptosis in damaged cells represents a defense mechanism mediated by the tumor suppressor p53 in response to diseases or noxious agents [28,29,30,31]. The aim of this research was to understand whether HOPS levels related to monoallelic deletion are either adequate to allow p53 stabilization or, conversely, result in apoptotic defects related to haploinsufficiency. In this study, we analyzed *Hops*^+/−^ mice and mouse embryonic fibroblasts (MEFs) to determine whether the heterozygous state was defective in DNA repair in response to etoposide-induced DNA damage. In particular, to dissect the complex relationship between HOPS and the tumor suppressor p53, we analyzed etoposide-induced apoptosis in responsive organs, such as thymus and spleen, of HOPS wild-type (*Hops*^+/+^), heterozygous (*Hops*^+/−^), and knock-out (*Hops*^−/−^) mice. Therefore, to analyze HOPS’ role in response to DNA damage, we generated three independent populations of immortalized murine embryonic fibroblasts bearing all three allelic arrangements for *Hops* gene.

We demonstrated that in vitro *Hops*^+/−^ and *Hops*^−/−^ MEFs and in vivo responsive organs from corresponding mice exhibited an impaired DNA-damage response to double-strand breaks (DSBs) induced by etoposide when compared to their wild-type counterparts. Defects in p53 stabilization, p53-mediated apoptosis, and DNA repair were observed. These data imply a novel role for HOPS, in which alteration seems to generate a defective control of DNA damage repair related to its haploinsufficiency.

## 2. Results

### 2.1. HOPS Expression and Characterization in Heterozygous Mice

In order to examine *Hops* heterozygous mouse gene expression, several tissues from 10 weeks old mice were collected to quantify baseline levels of HOPS protein. To characterize our model, HOPS amounts were measured in homozygous, heterozygous, and nullizygous mice. We first evaluated macroscopically the morphology of the explanted organs but, since the *Hops* gene appears not to be required for normal mouse development, we found no differences under basal conditions between the three genotypes (data not shown). However, as shown in Figure 1A, both wild-type and heterozygous mouse tissues expressed the protein and, obviously, no expression was revealed in knock-out mice samples. As expected, the heterozygous mice tissues exhibited lower amounts of HOPS compared to the wild type, while in the knock-out mouse tissues the protein was undetectable. We found a sharp reduction in HOPS protein between *Hops*^+/+^ and *Hops*^+/−^ genotypes in all the tissues examined, with peaks of diminution nearly 50% of the total (Figure 1B). 

Moreover, to delineate HOPS expression and define any possible defect in HOPS level, we measured the *Hops* mRNA basal transcript levels in spleen and thymus. In this case, heterozygous tissues presented higher *Hops* mRNA expression than the *Hops*^+/+^ counterpart, regardless of the amount of protein transcribed (Figure 1C). The data suggest that the heterozygous setting induced a compensative counteraction to the protein level drop by increasing the transcriptional machinery activity in order to sustain the cellular functions engaging HOPS.

### 2.2. Hops Heterozygous Mice Were Defective in DNA-Damage-Induced Repair

HOPS has been shown to play an interesting role in the regulation of p53 fate and functions [3,11]. High levels of activated and stabilized p53 protein accumulate in the nucleus in response to various cell stresses, including DNA damage [13,32,33]. Activated p53 can induce cell cycle arrest, DNA repair processes, and apoptosis. Following apoptogenic stimuli, HOPS binds p53, sustains p53 levels in the cytoplasm by reducing its ubiquitination, and ultimately contributes to the p53 mitochondrial apoptotic program [11]. In HOPS deficiency conditions, p53 expression is reduced along with DNA-damage-induced apoptosis [11]. In modulating p53 biology, HOPS exerts a potential role as tumor-suppressor protein. In order to investigate this possible feature, we analyzed the ability of *Hops* heterozygous models to stabilize p53 and respond to DNA damage insults. Ex vivo cells from thymus and spleen—organs known to be more responsive to apoptosis [34]—were treated with the topoisomerase II inhibitor etoposide at a concentration of 5 µM for 2, 4, and 6 h. Consistently with previous reports [11], Western blot analysis revealed a strong increase in p53 levels after treatment in *Hops*^+/+^ splenocytes and thymocytes, while their *Hops*^−/−^ counterparts failed to completely activate p53. In particular, wild-type cells showed a rapid increase in p53 protein level starting 2 h after treatment, while the knock-out ones exhibited a reduced and delayed p53 induction. Nevertheless, *Hops*^+/−^ samples revealed an intermediate pattern, more comparable to *Hops*^−/−^ than to *Hops*^+/+^ mice, with a weaker p53 signal throughout the treatment (Figure 2A).

In the same conditions, we inspected p53 transcriptional activation in transgenic mouse tissues. In all the genotypes (*Hops*^+/+^, *Hops*^+/−^, and *Hops*^−/−^) the etoposide treatment induced a progressive p53 transcription enhancement (Figure 2B). However, transcriptional activity in heterozygous splenocytes and thymocytes appeared to be deregulated compared to wild-type cells, with levels of p53 expression substantially higher in the heterozygotes at both basal and treated conditions. In response to DNA damage, reduction/lack of *Hops* induced genomic instability, which resulted in an increase of p53 transcript levels in both heterozygous and knock-out conditions (Figure 2B).

We next examined the levels of p21, a downstream effector and transcriptional target of p53. Indeed, in response to DNA damage, glucose deprivation, or irradiation [35], p21 levels increased in a p53-dependent manner and contributed to arrest cell proliferation [36,37]. Here, we showed that, similarly to p53, the p21 protein results increased after stress stimuli in both *Hops^+/+^* splenocytes and thymocytes (Figure 2A), with weaker immunoblotting signal in *Hops^+/−^* and *Hops^−/−^* tissue extracts. At the same time, p21 transcriptional levels in heterozygous splenocytes and thymocytes appeared to be boosted when compared to wild-type cells, in line with p53 transcription levels in response to DNA damage (Figure 2C).

### 2.3. Hops Heterozygous Mice Showed Defective Apoptosis Induction

As p53 is a key regulator of apoptosis [14,38,39], we were interested in determining possible differences in the apoptosis levels. Motivated by the idea that heterozygous mice are defective in the induction of p53 expression after DNA damage, we examined their ability to promote apoptosis following stress stimuli. Previous data from knock-out MEFs showed that lack of *Hops* resulted in remarkably reduced apoptosis levels, making cells more resistant to apoptosis [11]. To evaluate the functional role of *Hops* in mediating apoptotic effects in heterozygous mice, freshly explanted splenocytes and thymocytes were exposed to etoposide for 4 h and cell death was verified by flow cytometry analysis of permeabilized cells stained with annexin (Figure 3A,B). In line with previous results, following 4 h of treatment, *Hops*^−/−^ cells showed a drop in the percentage of apoptotic cells compared to *Hops*^+/+^ control. Notably, the analysis of *Hops*^+/−^ cells revealed a significant resistance to apoptosis following stress stimuli. We found a similar percentage of apoptotic cells both in *Hops*^−/−^ and *Hops*^+/−^ splenocytes and thymocytes in comparison to the wild-type cells. The data, consistent with previous Western blot analysis, confirmed a decreased p53 activity in cells from heterozygous mice which, similarly to *Hops*^−/−^ cells, results in reduced apoptotic outcome. Taken together, the results indicate that HOPS plays a pivotal role in regulating p53-mediated apoptosis, with a single monoallelic deletion responsible for the reduction of HOPS’ function as a p53 protein regulator.

### 2.4. The p53 Protein Showed a Shortened Half-Life in Hops Heterozygous MEFs

The p53 tumor suppressor is a short-lived protein that is stabilized and activated in response to a range of cellular stresses, including DNA damage [13,40,41]. In addition, our previous studies showed that HOPS controls p53 protein stability [11]. To dissect the complex relationship between HOPS and the tumor suppressor p53, we analyzed *Hops*^+/+^, *Hops*^+/−^, and *Hops*^−/−^ immortalized MEFs to determine whether they were defective in DNA repair in response to etoposide. We generated a single immortalized MEF clone for each *Hops* genotype, and we confirmed the corresponding genomic asset via PCR analysis and by assessing *Hops* mRNA and protein levels (Appendix A), before moving on to a specific experimental procedure. All clones showed p53 expression inducible by stresses and levels were detectable by two independent p53 antibodies specific to different epitopes. In order to validate our model, we examined HOPS’ ability to stabilize p53 in such MEFs by analyzing the kinetics of p53 turnover in a cycloheximide (CHX) assay (Figure 4). Our results confirmed that, as in *Hops*^−/−^ primary MEFs [11], p53 protein rapidly dropped in the *Hops*^−/−^ MEF clones. Furthermore, in the heterozygous MEFs compared to the wild-type control we observed that over the indicated time (0–120 min), p53 protein expression reproduced the *Hops*^−/−^ course, rapidly decreasing within 5 min of CHX treatment. The translated decay graph clearly indicates that p53 half-life in heterozygous MEFs was significantly reduced, even suggesting a destabilization of the p53 protein due to reduction of HOPS amount in the cells.

### 2.5. Hops Heterozygous MEFs Exhibited Altered p53 Activation Following DNA Damage

We previously investigated the role of *Hops* heterozygous cells in ex vivo response to etoposide. To further confirm our understanding of HOPS involvement in the differences of the cellular responses to DNA damage, MEFs were treated with etoposide (10 µM) for different lengths of time (0, 2, 4, 6, 8, 24 h) and analyzed for possible dysfunctional p53 activation. As expected, treatment of *Hops*^+/+^ MEFs with etoposide boosted p53 expression over time, as assessed by Western blot analysis (Figure 5A). Once again, we observed a sharp contrast between wild-type and heterozygous MEFs, with levels of p53 protein expression substantially higher in the *Hops*^+/+^ MEFs compared to *Hops*^+/−^ and *Hops*^−/−^ MEFs (Figure 5A). Moreover, we showed a difference in p53 phosphorylation at Ser15 among the three *Hops* genotypes. The kinetics of the response in terms of induction of p53 phosphorylation lagged in both *Hops*^+/−^ and *Hops*^−/−^ MEFs. While p53 phosphorylation activation was revealed within 2–4 h of treatment with etoposide, the peak of p53 activation was delayed in *Hops*^+/−^ and *Hops*^−/−^ MEFs, suggesting an alteration in triggering of the program of gene expression. Indeed, p53 phosphorylation at Ser15 is crucial for its stabilization and essential for the induction of cell cycle arrest and transcription of the p53-dependent apoptotic mediators [42,43,44,45]. We investigated the effect of etoposide on two representative transcriptional targets of p53 that play important roles in DNA-damage-induced apoptosis: the Bcl-2-associated X protein (Bax) and the BH3-only pro-apoptotic protein (PUMA) [46,47,48]. As expected, in *Hops^+/+^*, the two mediators showed a smooth but protracted increase in protein levels, correlating to the p53 activation. Conversely, in *Hops^+/−^* and *Hops^−/−^,* both Bax and PUMA presented a flattened response to etoposide treatment, with a higher baseline protein level (Figure 5A). At the transcriptional level, *Hops*^+/+^ MEF clones showed a time-dependent induction of p53-dependent transcripts over time (Figure 5B), while a short etoposide treatment (1.5–3 h) did not increase their levels in any of the clones (data not shown). Interestingly, the analyzed p53-downstream genes presented increased transcript levels in *Hops*^+/−^ MEFs compared to the *Hops*^+/+^ MEFs. *Bax* and *PUMA* expression appeared to be significantly increased at 18–24 h in *Hops*^+/−^ MEFs following etoposide treatment, while their expression in *Hops*^+/+^ cells was less pronounced (Figure 5B). It appears that p53-dependent gene expression is independent from the increased phosphorylation of p53, possibly suggesting that Ser15 phosphorylation is dispensable for p53 activity.

## 3. Discussion

The identification and characterization of novel genes involved in the regulation of biological processes such as cell division, differentiation, and stress response represents one of the most important aims in improving our understanding of the complex molecular mechanisms governing cellular oncogenesis. The tumor suppressor p53 oversees an extremely complex network within the living cell [33,49,50], playing a pivotal role in protecting cells from malignant transformation by inducing cell cycle arrest or apoptosis [15,32,41]. Several signals, including DNA damage, lead to the functional activation of p53, which becomes stabilized and accumulates in the nucleus where it induces the transcription of genes involved in these processes [13,30,40,49,51]. Recent data indicate a control of p53 stability mediated by the novel protein HOPS [11]. Our studies demonstrated that HOPS acts as a regulator of cytoplasmic p53 levels, promoting p53 recruitment to mitochondria and apoptosis induction [11]. Here, we aimed to investigate the functional interdependence between the p53 tumor suppressor and HOPS, considering the heterozygous state and potential outcomes of haploinsufficiency.

The ex vivo and in vitro analyses of *Hops* heterozygous mice and cells revealed a defective activation of p53 following a DNA-damaging stimulus such as etoposide. Notably, our study indicated that following stress stimulus in heterozygous mouse tissues, lower p53 protein is detected but higher gene expression. We thus propose that the increased transcription of HOPS and p53, proteins which strongly interact in both cytoplasm and mitochondria in wild-type conditions [11], might be necessary to contribute to p53 stabilization, which can fail at the post-translational level. Moreover, the significance of haploinsufficiency in heterozygous mice was underscored by the in vitro relevance of the cytofluorimetric analysis, revealing a subtle apoptotic phenotype in response to DSBs comparable to the *Hops*^−/−^ genotype. Loss of apoptotic response via repression of p53 functions appears to be required for malignant progression [52,53]. We thus report that decreased levels of HOPS in explanted cells resulted in reduced p53 activation and apoptosis. The monoallelic deletion was enough to destabilize the p53 protein and its functions, highlighting the role of *Hops* as a tumor suppressor. Indeed, the central role played by HOPS in controlling the stability and functions of important cellular players such as NPM, ARF, NF-κB [12], and p53 could account for a putative role of HOPS in cancer. Analysis of TCGA (the Cancer Genome Atlas) databases has shown that HOPS levels in cancer are altered in a high percentage of the tumor tissues examined, especially in colon and central nervous system neoplasms [54]. For these reasons, HOPS expression in cancer cells might be extremely important in correlation to p53 level [55].

The results presented in this study extend our previous findings on the role of HOPS in stabilizing the tumor suppressor p53 in response to DNA damage. We demonstrated that decreased levels of HOPS in *Hops* heterozygous mice and cells negatively modulated p53 signaling, leading to haploinsufficient effects possibly involving post-translational mechanisms affecting the binding between HOPS and p53. In conclusion, our results demonstrate for the first time in vitro and ex vivo that *Hops* hemizygous present haploinsufficiency features which lead to a reduction in p53 activation and alterations in RNA level for a number of key cellular p53-dependent genes involved in the DNA-damage response. Further studies should uncover the controversial discrepancy between *Hops* levels, p53, and the other upregulated genes found in *Hops* heterozygous MEFs. Moreover, in vivo experiments will improve our understanding of the HOPS–p53 interaction, clarifying whether the reduction in the HOPS-mediated p53 response observed in hemizygosity might lead to a different susceptibility to cancer. In the future, these results might be used in translational medicine for new human therapies and HOPS levels in cancer cells might be used to monitor and/or promote the activation of the p53 pathway.

## 4. Materials and Methods

### 4.1. Mice, Models, and Treatments

All experiments involving animals and their care were conducted in line with the guidelines of the University of Perugia Ethical Committee (Prot. n.54 30/05/2011) and the European Communities Council Directive 2010/63/EU. The mice were held in the animal facility of Perugia University and used for experiments. All the animals were maintained under specific defined flora/pathogen-free conditions in ventilated (high-efficiency particulate-arresting filtered air) sterile microisolator cages at a constant temperature (24–26 °C), constant humidity (30–50%) and a 12-h light/12 h dark cycle. Sterilized food and tap water were given ad libitum. 

*Hops*^−/−^ mice were obtained by homologous recombination [11]. The resulting chimera was mated to C57BL/6 females to generate heterozygous (*Hops*^+/−^) mice for the exon 2–3 deletion, while the heterozygous mice were intercrossed with homozygous (*Hops*^−/−^) counterparts, yielding *Hops*^−/−^ mice. 

For the morphologic analysis, 10 week old mice *Hops*^+/+^, *Hops*^+/−^, and *Hops*^−/−^ were sacrificed and the selected organs were analyzed for weight, dimension, and protein content. For the topoisomerase II inhibitor etoposide (Sigma-Aldrich, St. Louis, MO, USA) treatment, 4 to 5 week old male *Hops*^+/+^, *Hops*^+/−^, and *Hops*^−/−^ mice were sacrificed by cervical dislocation, the target organs (thymus and spleen) were homogenized, and the cells were immediately harvested for etoposide treatment (5 µM). Samples were then prepared for protein and RNA extraction.

### 4.2. Establishment of Immortalized MEFs

*Hops*^+/+^, *Hops*^+/−^, and *Hops*^−/−^ MEFs were prepared, respectively, from C57BL/6 mice and *Hops^−/−^* C57BL/6 mice, and the preparation protocol was adapted from Xu J. [56]. The pregnant mice were sacrificed at 13.5 d.p.c. (day post-coitum) by cervical dislocation as previously described [11]. The single clones were selected and immortalized according to the 3T3 assay [56]. The MEFs were cultured in Dulbecco’s modified Eagle’s medium (D-MEM, Sigma Aldrich, St. Louis, MO, Stati Uniti) containing 10% North American fetal bovine serum (NA-FBS, Corning, Glendale, AZ, USA), 1% penicillin–streptomycin, 1% glutamine (EuroClone, Milan, Italy), 1% nonessential amino acid (NEAA, EuroClone, Milan, Italy), and 250 μg/mL gentamicin (Gibco^®^, Invitrogen, Carlsbad, CA, USA) and maintained at 37 °C with 5% CO_2_.

### 4.3. RNA Extraction and Real-Time PCR

For quantitative PCR (qPCR), the RNA was extracted from tissues and cell lines using NucleoZOL reagent (Macherey-Nagel, Düren, Germany) according to the manufacturer’s instructions. Total RNA (1 µg) was reverse transcribed using an iScript cDNA Synthesis kit (Bio-Rad, Hercules, CA, USA) with random primers for cDNA synthesis. qPCR was performed with SYBR Green qPCR Master Mix (Thermo Fisher Scientific, Waltham, MA, USA) using the QuantiStudio3 Real-Time PCR System (Applied Biosystems, Foster City, CA, USA). The expression of all target genes was validated and normalized relative to β-actin expression using the 2^−ΔΔCt^ method. The primers used are listed in Appendix A.

### 4.4. Western Blot Analysis and Antibodies

The total proteins were collected from organs or immortalized MEFs from *Hops*^+/+^, *Hops*^+/−^, and *Hops*^−/−^ mice. Protein extracts were denatured in Laemmli buffer (Tris/HCl at pH 6.8, 200 mM, SDS 8%, bromophenol blue 0.4%, glycerol 40%, and b-mercaptoethanol 5%) and boiled for 5 min at 95 °C. Protein extracts were normalized by SDS–PAGE and Coomassie blue (blue bromophenol solution 0.1%, acetic acid 15%, and methanol 25%) staining. Proteins were separated on polyacrylamide gel and transferred by electroblotting onto nitrocellulose membranes (Bio-Rad, Hercules, CA, USA). The membranes were blocked in dry fat-free milk 5% in PBS and probed overnight with the following primary antibody: anti-p53 (clone A1 sc-393031, Santa Cruz Biotechnology, Dallas, TX, USA), anti phospho-p53 (Ser15) (#9284, Cell Signaling Technology, Danvers, MA, USA), β-actin (A2066, Sigma-Aldrich, St. Louis, MO, Stati Uniti). For the detection of mouse HOPS protein, a rabbit polyclonal antibody produced as previously described was used [11]. Primary antibodies were revealed by anti-rabbit/anti-mouse HRP-conjugated secondary antibodies (Bio-Rad, Hercules, CA, USA) and detection was achieved with ECL (Clarity Max ECL, Bio-Rad, Hercules, CA, USA). Signal images of each protein were acquired using the Bio-Rad ChemiDoc™ Imagers camera. The protein densities were semi-quantified using the Fiji/ImageJ Software.

### 4.5. Flow Cytometry Analysis

The apoptosis induced in *Hops*^+/+^, *Hops*^+/−^, and *Hops*^−/−^ splenocytes and thymocytes after treatment with etoposide (5 µM) for 2–4 h was measured by flow cytometry. The cells were collected by centrifugation, washed with PBS, stained with Fixable Viability Dye eFluor^®^ 780 (eBioscience, San Diego, CA, USA) for 30 min, and then washed with PBS and Flow Cytometry Staining Buffer. Annexin V conjugated with PerCP-eFluor 710 (eBioscience, San Diego, CA, USA) was added and incubated with cells at room temperature for 15 min. The percentage of positive apoptotic cells was determined by flow cytometry.

### 4.6. Statistical Analysis

All statistical analyses were performed using Prism 8.0 (GraphPad, La Jolla, CA, USA). Data were collected for the control group and the treated group. Each experiment was performed at least three times. The data are presented as mean ± SEM. The significance test of all the data was analyzed with either two-way ANOVA Turkey’s multiple-comparison test or ordinary one-way ANOVA Dunnett’s multiple comparison test. *p*-values less than 0.05 were considered significant: * *p* < 0.05, ** *p* < 0.01, *** *p* < 0.001 and **** *p* < 0.0001

## Figures and Tables

**Figure 1 ijms-22-07186-f001:**
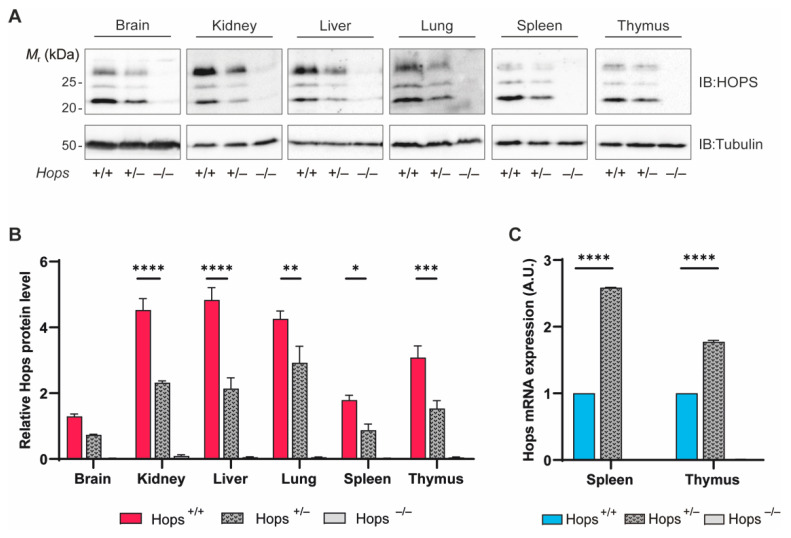
Baseline level of HOPS in *Hops*^+/+^, *Hops*^+/−^, and *Hops*^−/−^ mice. (**A**) Protein extracts from *Hops*^+/+^, *Hops*^+/−^, and *Hops*^−/−^ mouse tissues were analyzed by Western blot using anti-HOPS antibody and anti-α-tubulin antibody as the loading control. (**B**) Densitometric analysis showed the average protein level of HOPS after normalization to the housekeeping protein from at least three independent experiments. Representative blots are shown. (**C**) HOPS mRNA levels were quantified by RT-PCR in *Hops*^+/+^, *Hops*^+/−^, and *Hops*^−/−^ thymus and spleen. β-actin was used as a housekeeping gene, and relative *Hops* levels in wild-type mice tissues were assumed to be 1. Data were analyzed using two-way ANOVA with Tukey’s multiple-comparisons test. Values represent mean ± SEM. * *p* < 0.05, ** *p* < 0.01, *** *p* < 0.001, and **** *p* < 0.0001.

**Figure 2 ijms-22-07186-f002:**
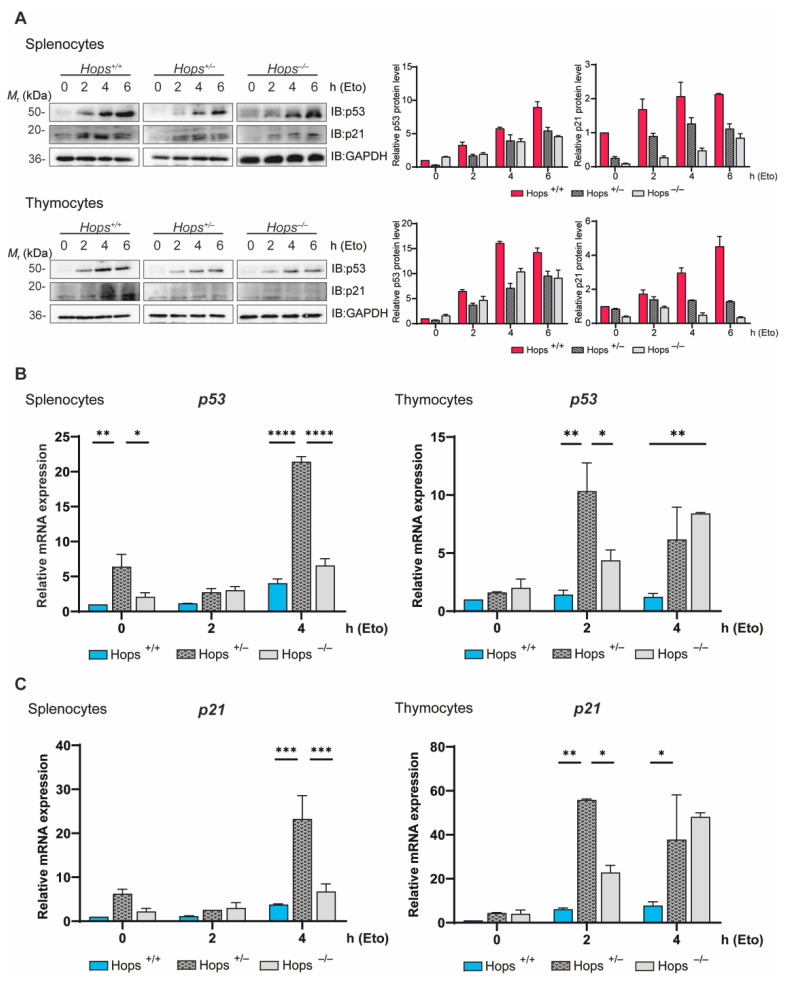
Ex vivo analysis from thymus and spleen after etoposide treatment. (**A**) Western blot analysis of p53 and p21 tumor-suppressor protein amounts in splenocytes (upper panel) and thymocytes (lower panel) from *Hops*^+/+^, *Hops*^+/−^, and *Hops*^−/−^ mice following etoposide (Eto) treatment at the indicated times (h). GAPDH antibody was used as the loading control. Graphical quantification of Western blots is shown in the right panel. The experiment was performed three times in explanted cells from four different mice. Representative panels are shown. The (**B**) p53 and (**C**) p21 mRNA levels were analyzed by RT-PCR in ex vivo splenocyte and thymocyte cells following etoposide treatment at the indicated times. Data (mean ± SEM of three independent experiments) are represented as normalized transcript expression to β-actin. The *p*-values were calculated by two-way ANOVA followed by Tukey’s multiple-comparisons test. * *p* < 0.05, ** *p* < 0.01, *** *p* < 0.001, and **** *p* < 0.0001.

**Figure 3 ijms-22-07186-f003:**
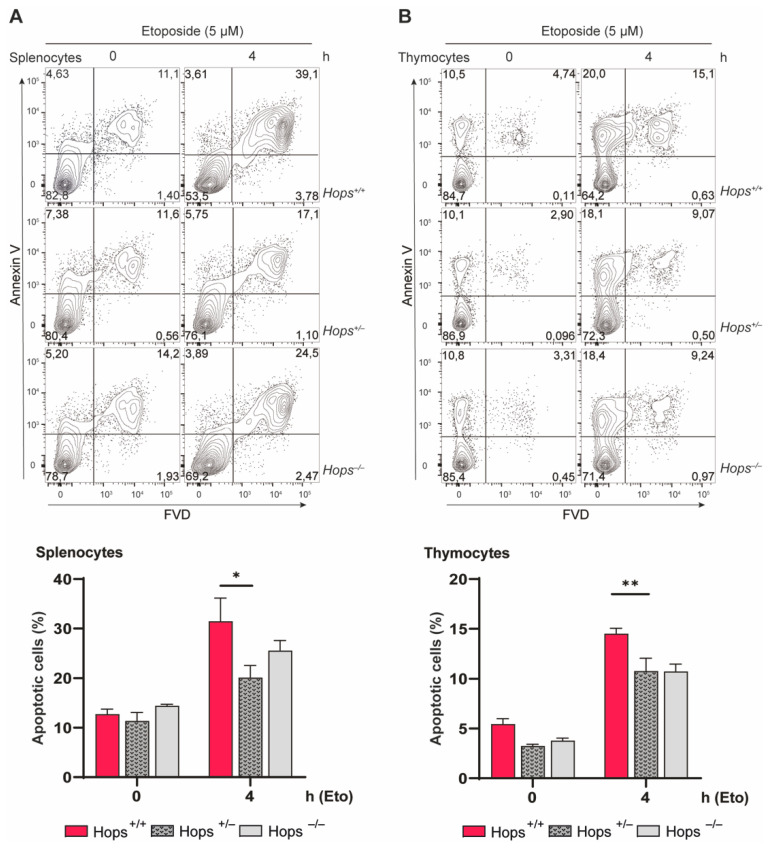
Apoptosis induction in heterozygous mice. (**A**) Splenocytes and (**B**) thymocytes from *Hops*^+/+^, *Hops*^+/−^, and *Hops*^−/−^ mice were co-cultured with etoposide at 5 μM for 4 h, and apoptosis was analyzed by flow cytometry. Representative cytofluorimetric dot plots (upper panels) and relative histograms of percent expression (lower panels) were represented. Data represent mean ± SEM and were analyzed using two-way ANOVA with Tukey’s multiple-comparisons test. * *p* < 0.05, ** *p* < 0.01.

**Figure 4 ijms-22-07186-f004:**
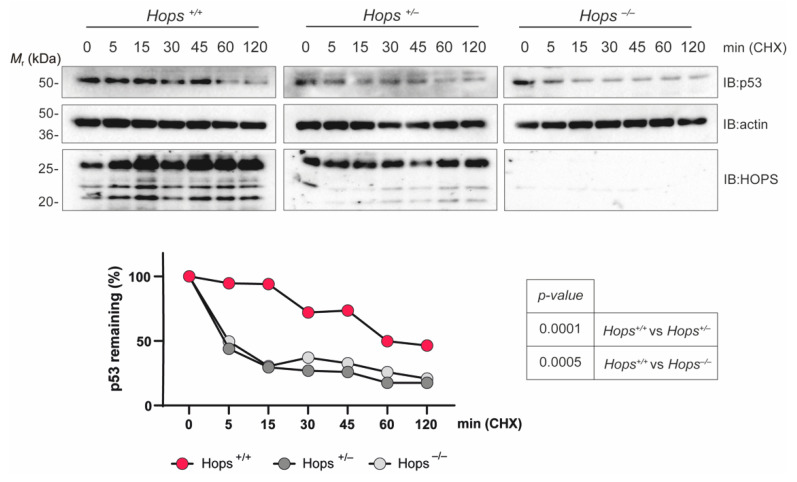
p53 protein stability in MEFs. *Hops*^+/+^, *Hops*^+/−^, and *Hops*^−/−^ MEFs were subjected to cycloheximide (CHX) treatment. Cycloheximide was added at 100 μM for the indicated time (h) and p53 survival was analyzed by Western blot analysis (upper panel) and shown as a graph (lower panel). Representative images of three experiments were shown. The p53 protein levels were semiquantified using β-actin as a loading control, and relative p53 levels at time 0 were assumed to be 100% (lower panel). The *p*-values were calculated by two-way ANOVA followed by Tukey’s multiple-comparisons test and are represented in the table.

**Figure 5 ijms-22-07186-f005:**
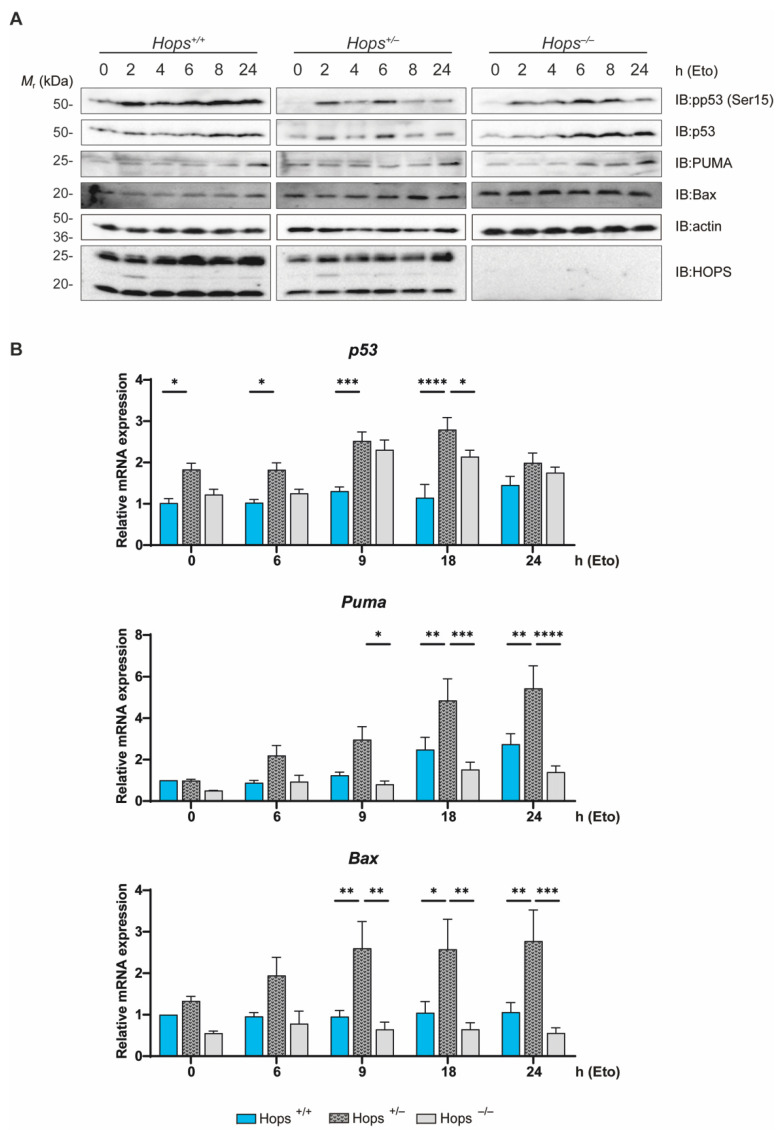
p53 transcriptional and non-transcriptional activation upon etoposide treatment. (**A**) Western blot analysis in *Hops*^+/+^, *Hops*^+/−^, and *Hops*^−/−^ MEFs of p53 tumor-suppressor protein amount, phosphorylation at Ser15 (p-p53 (Ser15)), Bax, and PUMA, following treatment at the indicated times. β-actin antibody was used as the loading control. The experiment was performed three times and representative panels are shown. (**B**) p53, PUMA, and BAX mRNA levels were analyzed by RT-PCR in cells treated as in A. The *p*-values were calculated by two-way ANOVA followed by Tukey’s multiple-comparisons test. * *p* < 0.05, ** *p* < 0.01, *** *p* < 0.001, and **** *p* < 0.0001.

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
