# Peer review of "Hops/Tmub1 Heterozygous Mouse Shows Haploinsufficiency Effect in Influencing p53-Mediated Apoptosis"

_ijms, 2021, doi:10.3390/ijms22137186_

Round 1

Reviewer 1 Report

In the present manuscript by Ferracchiato et al., the Authors are describing the effect of the heterozygous loss of a ubiquitin like protein, HOPS.

HOPS has been already demonstrated to increase the stability of p53 and the current work aims at describing its effect on p53 transcriptional activity in particular in response to DNA damage.

I personally think this manuscript is conceptually well organised and easy to read. The conclusions are sustained by the experiments and the manuscript might be potentially interesting for the broad audience of IJMS.

I personally think most of my doubts are referred to the lack of evidence at a protein level, since many experiments rely just on mRNA expression. Therefore, I just have a few minor remarks or suggestions for the Authors.

- Figure 2 A should show bar charts. Moreover the number of replicates should be indicated in the figure legends.

- in Figure 2 western blot analysis of p21 should be shown.

- Please show bar charts on the graph of figure 4.

- The Authors should show western blot of Puma and Bax in Figure 5.

Finally, I have a few curiosities that the Authors should maybe include in the discussion section, if data are available:

- is it known the behaviour of HOPS in the context of mutated p53?

- what about HOPS expression in cancer? Is there any correlation with p53 expression and/or with p53 mutants expression?

Author Response

Here, you will find the rebuttal point-by-point letter to the reviewers, which with their criticism have improved our manuscript

Reviewer 2 Report

In this manuscript, Simona Ferracchiato et al showed that HOPS and p53 's relationship. Also, HOPS levels on the DNA damage response by examining transcript profiles of p53-dependent genes, showing a suggestive deregula-
tion of the mRNA levels for a number of p53-dependent genes.

These findings are potentially interesting. The manuscript could be further strengthened with a few additional experiments denoted below.

  1. The authors need to explain about p53 and apoptosis in introduction part.
  2. In Figure 2A, authors need to show the p21 (p53 target gene) expression by western blotting.
  3. In Figure 3, authors should explain why apoptotic cells are not that much in Hops+/- compare with Figure 2A and B data (p53 expression) because the expression of p53 is very high, while apoptosis is less likely.
  4. It would be more significant if authors should show the apoptotic proteins such as PARP, Bcl-2, Bcl-xL etc by western blotting. 
  5. There are several places that incorrectly or inaccurately write down the manuscript. Authors need to pay close attention to proper labeling of this manuscript.

Author Response

Here, you will find the rebuttal point-by-point letter to the reviewers, which with their criticism have improved our manuscript.
